# DNA Repair Inhibitors Potentiate Fractionated Radiotherapy More Than Single-Dose Radiotherapy in Breast Cancer Cells

**DOI:** 10.3390/cancers14153794

**Published:** 2022-08-04

**Authors:** Wen-Kyle Wong, Francisco D. C. Guerra Liberal, Stephen J. McMahon

**Affiliations:** The Patrick G Johnston Centre for Cancer Research, Queen’s University Belfast, 97 Lisburn Road, Belfast BT9 7AE, UK

**Keywords:** fractionated radiotherapy, sublethal damage repair, radiosensitization, DDR inhibition, ATM, ATR, PARP, breast cancer

## Abstract

**Simple Summary:**

DNA damage response (DDR) inhibitors have been shown to sensitize cells to radiation yet have seen limited application in clinical settings. This could be due to a lack of understanding of how these inhibitors interact with ionizing radiation (IR) dose fractionation and cellular repair. Our study investigated the radiosensitizing effect of different DDR inhibitors on human breast cancer cells, utilizing single-dose and fractionated IR. Their effect on damage repair, DNA double-strand break repair kinetics and cell cycle distribution was also evaluated. The main finding was that radiosensitization by DDR inhibition was more prominent when combined with fractionated IR than single-dose IR. Moreover, DDR inhibition impeded the repair of IR-induced DNA double-strand breaks. Altogether, our study established the radiosensitizing potential of DDR inhibitors while highlighting the importance of IR dose fractionation in similar studies.

**Abstract:**

Pharmacological inhibitors of DNA damage response (DDR) proteins, such as the ataxia-telangiectasia mutated (ATM) and ataxia-telangiectasia and Rad3-related (ATR) kinases and poly (ADP-ribose) polymerase (PARP), have been developed to overcome tumor radioresistance. Despite demonstrating radiosensitization preclinically, they have performed suboptimally in clinical trials, possibly due to an incomplete understanding of the influence of DDR inhibition on ionizing radiation (IR) dose fractionation and sublethal damage repair. Hence, this study aimed to evaluate the radiosensitizing ability under fractionation of ATM inhibitor AZD0156, ATR inhibitor AZD6738 and PARP inhibitor AZD2281 (olaparib), utilizing MDA-MB-231 and MCF-7 human breast cancer cells. Clonogenic assays were performed to assess cell survival and sublethal damage repair after treatment with DDR inhibitors and either single-dose or fractionated IR. Immunofluorescence microscopy was utilized to evaluate DNA double-strand break repair kinetics. Cell cycle distributions were investigated using flow cytometry. All inhibitors showed significant radiosensitization, which was significantly greater following fractionated IR than single-dose IR. They also led to more unrepaired DNA double-strand breaks at 24 h post-IR. This study provides preclinical evidence for the role of AZD0156, AZD6738 and olaparib as radiosensitizing agents. Still, it highlights the need to evaluate these drugs in fractionated settings mirroring clinical practice to optimize the trial design.

## 1. Introduction

Targeting the DNA of tumor cells is a major area of research in cancer therapeutics with strategies mainly involving two approaches: inducing DNA damage and inhibiting the DNA damage response (DDR). The former can be achieved with radiotherapy, with its role in cancer management being well established for over a century [1]; the latter involves designing drugs to inhibit DNA repair effectors and its emergence as a therapeutic strategy is relatively recent [2]. Given that they have a common end goal of generating lethality via genomic instability, their simultaneous administration in treatment regimens is promising, especially since DDR over-activation has been suggested as a mechanism of radioresistance [3]. However, the interactions between radiotherapy and DDR inhibitors require further investigation to justify their utility as combination therapies.

The DDR consists of a myriad of events triggered through multiple pathways to minimize the potentially catastrophic effects of DNA lesions [4]. Given the vital role of DNA in living cells as well as its susceptibility to various sources of damage due to its large intricate structure, the protective mechanisms required to ensure genome stability are unsurprisingly complex and involve interactions between different processes. If the DNA damage encountered is deemed irreversible, the DDR ultimately results in cell death, preventing mutagenic effects such as tumorigenesis [5]. Conversely, sublethal DNA damage can be reversed via the DDR, with the mechanisms divided into the single-strand break (SSB) and double-strand break (DSB) repair, as determined by the extent of sustained lesions (Figure 1).

Other than repairing strand breaks, the DDR activates indirect but nonetheless crucial processes for restoring DNA integrity, for instance, cell cycle arrest. As DNA repair requires time to take place, the DDR can facilitate its completion by temporarily halting the cell cycle. This is mainly achieved by inhibiting cyclin-dependent kinases. Their phosphorylation of multiple downstream proteins is key for cell cycle progression through four specific checkpoints: G_1_/S, intra-S, early G_2_ and late G_2_ [6]. The G_1_/S checkpoint is particularly important as it controls the commencement of DNA replication. Another point of emphasis regarding the cell cycle in DDR is that certain DNA repair mechanisms are phase-specific. For example, homologous recombination (HR) can only occur during and after the S phase due to its requirement of sister chromatids.

Governing these diverse effects of the DDR are proteins from various families, and their roles can be described as damage sensors, signal transducers or effectors [6]. Different sensor proteins respond to distinct DNA lesions, with an example being the recognition of DSBs by the Mre11–Rad50–Nbs1 complex [7]. Subsequent activation of effector proteins then leads to the initiation of pathways that culminate in DNA repair, cell cycle arrest or apoptosis. The key intermediate step which connects damage sensing and effector pathways is the recruitment of signal transducers, as their capacity to modify downstream proteins enables the continuation of the DDR [8]. Important signal transducers include the kinases ataxia-telangiectasia mutated (ATM) and ataxia-telangiectasia and Rad3-related (ATR) as well as the poly (ADP-ribose) polymerase (PARP) enzyme family [9,10,11]. They are considered the main therapeutic targets regarding DDR inhibition due to their vast range of actions (Figure 2).

The finding that DDR activation is one of the main factors contributing to radioresistance lends support to the use of DDR inhibitors as radiosensitizing agents [12]. Numerous drugs have been designed against ATM, ATR, and PARP, including AZD0156, AZD6738, and olaparib. Preclinical studies have demonstrated their radiosensitizing effects [13,14,15], providing an incentive for their investigation in clinical trials. However, a phase II trial assessed veliparib, another PARP inhibitor, combined with whole-brain radiotherapy to treat metastatic brain disease in non-small cell lung cancer patients. Results indicated no significant improvement in overall survival [16]. This suggests further exploration of the synergy between radiotherapy and DDR inhibitors is required to improve the clinical translation.

Understanding the interaction between ionizing radiation (IR) treatment schedule and DDR inhibition could be vital in optimizing this novel combination therapy, considering the prominent role of fractionated radiotherapy in modern clinical practice [17]. In fractionated radiotherapy regimens, damaged but still surviving cells can activate their DDR mechanisms to repair DNA lesions induced by the initial dose. As a result, fewer sublethal lesions can interact with the next IR dose, thus reducing the overall response. While this can be used to optimize the relative response of tumor and normal tissue, high sublethal damage (SLD) repair capacity in cancer cells may lead to radioresistance. Therefore, it is important to understand if the addition of DDR inhibitors may abolish the ability of tumor cells to recuperate between fractions. Despite the importance of fractionation, preclinical studies directly comparing the effects of single-dose radiotherapy with fractionated radiotherapy in cells treated with DDR inhibitors are lacking. This represents a significant gap in the literature.

This study aims to improve our understanding of the interactions between radiotherapy and DDR inhibitors, targeting ATM (AZD0156), ATR (AZD6738), and PARP (olaparib). The radiosensitizing capability of these drugs was evaluated in vitro for both acute and fractionated exposures, investigating both overall response and SLD repair capacity. The influence of DDR inhibition on DNA DSB repair kinetics and cell cycle distribution was also assessed to obtain further mechanistic information regarding each drug. Breast cancer cell lines (MDA-MB-231 and MCF-7) were utilized because of the prevalent role of radiotherapy in breast cancer management. Furthermore, the occurrence of radioresistance, particularly in the aggressive triple-negative subtype, represents a clinical need that could potentially be met with the simultaneous use of DDR inhibitors in radiotherapy regimens [18].

## 2. Materials and Methods

### 2.1. Cell Lines

Human breast cancer cell lines MDA-MB-231 and MCF-7 were used in this study to encompass different molecular subtypes. MDA-MB-231 cells represent triple-negative breast cancer, whereas MCF-7 cells represent hormone receptor-positive breast cancer. Both cell lines were purchased from American Type Culture Collection (Manassas, VA, USA). Cells were propagated in DMEM medium supplemented with 10% fetal bovine serum and 1% Penicillin-Streptomycin and incubated at humidified 37 °C in 5% CO_2_. Cells used for all experiments were in logarithmic phase growth.

### 2.2. Drugs and Irradiation

AZD0156 and AZD6738 were used to target ATM and ATR, respectively, and both drugs were purchased from AstraZeneca plc (Cambridge, UK). Olaparib was purchased from Activate Scientific (Regensburg, Germany) and used for PARP inhibition. The specificity of AZD0156, AZD6738, and olaparib for inhibiting the actions of ATM, ATR, and PARP, respectively, has been previously demonstrated in other studies [13,14,19], which were used to guide the initial experimental design.

All three drugs were first prepared as 10 mM stock solutions in dimethyl sulfoxide (DMSO), then the required working concentrations were obtained by serial dilution with media. Drug treatment was conducted by replacing the culture media with treated media. DMSO was used as the mock treatment for control groups in drug-only experiments, with the DMSO concentration matching that of the highest treatment group. In experiments utilizing combinations of radiotherapy and DDR inhibitors, unirradiated controls were treated with DDR inhibitors only to account for drug-only cytotoxicity. All combination studies treated cells with the drugs one hour before irradiation. In experiments involving combinations of fractionated IR and DDR inhibitors, cells were treated with only a single drug dose before the first IR exposure. Cells were exposed to the drugs for the indicated durations or until they were assayed, without replacing the drugged media with fresh culture media. The concentrations of AZD0156 (0.01 µM), AZD6738 (0.25 µM) and olaparib (1 µM) used in all combination experiments were chosen based on their estimated IC_50_ in MDA-MB-231 and MCF-7 cells (see Section 3.2 later) as well as a review of similar studies [14,20,21], with the intention to minimize drug-only cytotoxicity while adequately eliciting the inhibitory effects of each DDR inhibitor on their protein targets.

For IR experiments, irradiation was carried out using an X-Rad 225 cabinet X-ray irradiator (Precision X-ray, Inc., North Branford, CT, USA), operated at a 225 kV peak, 13.3 mA and 50 cm shelf position with a dose rate of 0.591 Gy/minute. Exposure times were adjusted to obtain target IR doses. Cells were irradiated with a single dose or two equal doses at varying intervals to mimic fractionated radiotherapy from 15 min to 24 h.

### 2.3. Clonogenic Assays

The clonogenic assay was used to assess cell survival after treatment with radiotherapy, DDR inhibitors or combination therapies. It is considered the gold standard method for radiosensitivity measurement as it enables the determination of reproductive cell death, which is an integral effect of radiotherapy [22,23]. To begin, cells were seeded into 6-well plates using suitable seeding densities. Cells were then left in the incubator overnight to adhere before they were treated as required, either with IR, DDR inhibitors, or combinations of both. Subsequently, cells were returned to the incubator for 5–10 days to form colonies. Cells were fixed and stained using 0.4% *w*/*v* crystal violet in 90% industrial methylated spirit diluted with water. Only colonies with 50 cells or more were scored, with their visualization aided by a stereomicroscope and lightbox. The plating efficiency was determined from the control groups as below:$$\mathrm{Plating}\;\mathrm{efficiency}=\frac{\mathrm{Number}\;\mathrm{of}\;\mathrm{colonies}\;\mathrm{formed}}{\mathrm{Number}\;\mathrm{of}\;\mathrm{cells}\;\mathrm{seeded}}$$

The surviving fraction of the treatment groups was calculated as below:$$\mathrm{Surviving}\;\mathrm{fraction}=\frac{\mathrm{Number}\;\mathrm{of}\;\mathrm{colonies}\;\mathrm{formed}}{\mathrm{Number}\;\mathrm{of}\;\mathrm{cells}\;\mathrm{seeded}\;\times \;\mathrm{Plating}\;\mathrm{efficiency}}$$

All clonogenic assay data were obtained from three independent experiments consisting of technical duplicates or triplicates.

The mean surviving fractions from experiments involving radiotherapy were fitted to cell survival curves in Prism 8.4.3 (GraphPad Software, Inc., San Diego, CA, USA), using the linear-quadratic model as defined below:$$S=e^{-\alpha D-\beta D^{2}}\;$$
where *S* is the mean surviving fraction after treatment, *α* and *β* are the linear and quadratic parameters of intrinsic radiosensitivity, respectively, and *D* is the total IR dose administered in Gy. The different IR combination treatments were then compared by calculating the mean inactivation dose ($\bar{D}$), defined as the area under the survival curve, and the sensitizer enhancement ratio (SER), calculated by dividing the $\bar{D}$ for the IR-only groups with the IR and drug combination groups [24].

### 2.4. Drug IC_50_ Estimation

To estimate the IC_50_ of AZD0156, AZD6738 and olaparib in MDA-MB-231 and MCF-7 cells, treatment with each DDR inhibitor was conducted using drug concentrations ranging from 0.001 to 5 µM. Drugged cells were incubated for 5–10 days to allow colony formation. Cell survival after treatment was quantified with clonogenic assays as described in Section 2.3. Surviving fractions were normalized to DMSO control groups, with the mean and standard deviation (SD) calculated from three independent experiments. Finally, Prism 8.4.3 was used to plot three-parameter dose-response curves to obtain estimated drug IC_50_ values in both cell lines.

### 2.5. Immunofluorescence Microscopy of 53BP1 Foci

Immunofluorescence microscopy of 53BP1 foci was conducted to quantify DNA damage, specifically DSBs [25]. First, glass coverslips were sterilized with 70% ethanol. Fifty thousand cells were seeded on each coverslip and returned to the incubator overnight to adhere before being treated as indicated. At time points of 0.5, 2, 6 and 24 h after treatment, culture media were removed, and cells were washed with phosphate-buffered saline (PBS). Next, cells were fixed in a −20 °C solution of equal parts acetone and methanol for 10 min. Cells were permeabilized by incubating in 0.5% Triton X-100 (Sigma-Aldrich; Merck, Darmstadt, Germany, catalog number: T8787) in PBS for 20 min, followed by an hour-long incubation in a blocking buffer (0.1% Triton X-100, 5% fetal bovine serum in PBS) to prevent non-specific binding of antibodies. Subsequently, cells were incubated in 1:5000 53BP1 rabbit polyclonal IgG primary antibody (Novus Biologicals, LLC, Littleton, CO, USA, catalog number: NB100-304SS) diluted with blocking buffer for an hour. After draining and washing three times with PBS for five minutes each, cells were incubated in 1:2000 Alexa Fluor 568 goat anti-rabbit IgG (H + L) cross-adsorbed secondary antibody (Invitrogen; Thermo Fisher Scientific, Inc., Waltham, MA, USA, catalog number: A-11036) diluted with blocking buffer for an hour in the dark. Samples were washed in the dark three times with PBS for five minutes each. Finally, glass coverslips were mounted onto microscope slides using ProLong Gold Antifade Mountant with DAPI (Thermo Fisher Scientific Inc., Waltham, MA, USA, catalog number: P36931). Foci were manually counted with the ZEISS Apotome fluorescence microscopy system (Oberkochen, Germany) using an ×63 objective, and the number of 53BP1 foci was recorded in 50 cells from each sample. Data were normalized by subtracting the number of foci in untreated control groups. All immunofluorescence data were expressed as the mean and SD of at least two independent experiments. For repair kinetics analysis, foci data were fit with an exponential decay in Prism 8.4.3 as defined below:*N* = (*N*_0_ − *Plateau*)e^−*kt*^ + *Plateau*
where *N*_0_ represents the initial number of 53BP1 foci, *Plateau* represents the residual damage and *k* is the rate of DSB repair.

### 2.6. Western Blot

An hour before irradiation with 4 Gy, cells were treated with 0.1 µM AZD0156, 0.25 µM AZD6738 or 1 µM olaparib. Control cells were treated with DMSO. Then, cells were harvested 30 min after irradiation, and proteins were extracted with a radioimmunoprecipitation assay buffer. Then, 50 μg of protein sample was loaded onto a 6% SDS-PAGE or 8% SDS-PAGE gel, and after electrophoresis, proteins were blotted on a nitrocellulose membrane (Life Technologies, Carlsbad, CA, USA). The membranes were blocked with 3% bovine serum albumin (BSA) in Tris-buffered saline (TBS)-Tween (0.1% Tween-20 in TBS) and incubated overnight at 4 °C with primary antibodies (PAR(ab-1) (Milipore), pATR #2853, pATM #13050 (Cell Signaling, Danvers, MA, USA) at a dilution of 1:1000 in 3% BSA in PBS-Tween. The anti-Vinculin #13901 (Cell Signaling, USA) antibodies were used as housekeeping controls at a dilution of 1:1000. After washing with TBS-Tween, membranes were incubated in their secondary anti-rabbit and anti-mouse horseradish peroxidase-conjugated antibodies diluted at 1:2000 at room temperature for one hour. The membranes were then washed and developed with Luminata Crescedo Western HRP subtract (Millipore, Burlington, MA, USA) using the GBox Imager by Syngene (Cambridge, UK). Band density quantification was performed using the ImageJ software. Protein ratios were normalized to the untreated control cells.

### 2.7. Cell Cycle Analysis by Flow Cytometry

Flow cytometry was used to conduct cell cycle analysis based on the distribution of cellular DNA content. First, 150,000 to 300,000 cells were seeded onto culture dishes and incubated overnight to adhere before being treated as indicated. At time points of 1, 6 and 24 h after treatment, cells were harvested before being fixed in ice-cold ethanol. Fixed cells were kept in a −20 °C freezer at least overnight until they were stained for flow cytometry. The fixed cells were centrifuged at 700× *g* for 5 min at 4 °C, and the excess ethanol was removed. Cell pellets were then re-suspended in PBS and centrifuged again at 700× *g* for 5 min at 4 °C. Cell pellets were re-suspended in 1% propidium iodide and 0.25% RNase A (QIAGEN, Hilden, Germany, catalog number: 19101), used to ensure DNA-specific staining, in PBS. After incubating overnight at 4 °C, cell cycle analysis of the stained samples was conducted using the BD Accuri C6 Plus flow cytometer with the BD CSampler Plus software (BD Biosciences, San Jose, CA, USA). All cell cycle analysis data were obtained from three independent experiments.

### 2.8. Statistical Analysis

Statistical analysis and graphs were generated with Prism 8.4.3. Data were expressed as the mean ± SD from three independent experiments conducted in duplicate or triplicate unless otherwise stated. Unpaired two-sample *t*-tests were used to determine statistical differences between the means of the two groups. Ordinary one-way analysis of variance (ANOVA) with Dunnett multiple comparison tests or two-way ANOVA was performed for comparisons involving three or more groups. For all statistical tests, *p* values of less than 0.05 were considered statistically significant. Data for all figures are available in the Appendix A.

## 3. Results

### 3.1. IR Dose-Response Characterization of MDA-MB-231 and MCF-7 Cells

Clonogenic assays were performed to determine cell survival after irradiation to evaluate the effect of single-dose and fractionated IR on MDA-MB-231 and MCF-7 cells. Cells were exposed to total doses of 0.5 to 10 Gy, given in either a single dose or two equal doses 24 h apart. IR exposure reduced the colony-forming ability of the cells in a dose-dependent manner (Figure 3). Both cell lines demonstrated higher survival after 24-h fractionated compared to single-dose IR at all total IR doses, with the increase in survival due to dose fractionation being more evident at the larger total doses (Figure 3A,B). Both single-dose and 24-h fractionated IR inhibited the proliferation of MCF-7 cells more than MDA-MB-231 cells. While differences were not statistically significant at most individual dose points, two-way ANOVA indicated a significant difference in overall sensitivity between the two cell lines across the dataset as a whole (*p* = 0.001; Figure 3C,D).

The effect of IR dose fractionation was further investigated by evaluating SLD repair in irradiated MDA-MB-231 and MCF-7 cells. Two IR doses of 3 Gy were administered, separated by 0 to 4 h, and clonogenic assays were then used to determine cell survival. Both cell lines demonstrated higher survival as the interval between IR doses increased. Their SLD repair kinetics were similar, as indicated by their comparable repair half-times of approximately half an hour (Figure 4).

### 3.2. IC_50_ Estimation of AZD0156, AZD6738 and Olaparib in MDA-MB-231 and MCF-7 Cells

The drug-only cytotoxicity of AZD0156, AZD6738 and olaparib in MDA-MB-231 and MCF-7 cells was assessed with clonogenic assays, using drug concentrations of 0.001 to 5 µM. Estimated drug IC_50_ values are listed in Table 1. MDA-MB-231 cells were most sensitive to AZD0156, whereas MCF-7 cells were most sensitive to AZD6738. Olaparib was the least potent in both cell lines. Overall, the highest potency was achieved with AZD0156 in MDA-MB-231 cells, while this drug was approximately 22 times less potent in MCF-7 cells (*p* < 0.001).

### 3.3. AZD0156 Enhances Lethality of Single-Dose IR

The radiosensitizing capability of AZD0156 (0.01 µM), AZD6738 (0.25 µM) and olaparib (1 µM) was initially evaluated through their combined use with single-dose IR. MDA-MB-231 and MCF-7 cells were treated with each drug an hour before single-dose irradiation at 0.5 to 4 Gy, and clonogenic assays were performed to assess cell survival. The only DDR inhibitor which induced significant radiosensitization in these conditions was AZD0156. This enhancement of single-dose IR cytotoxicity was observed at all IR doses in MDA-MB-231 cells, whereas in MCF-7 cells, it reached significance only at 4 Gy. On the other hand, AZD6738 and olaparib did not lead to significant radiosensitization in either cell line (Figure 5A,B).

To establish whether the induced radiosensitization was due to additive or synergistic effects, observed surviving fractions were compared to expected surviving fractions after single-dose IR and DDR inhibitor combination treatments. The expected surviving fractions from each combination treatment were calculated by multiplying the observed surviving fractions after IR-only and drug-only treatments, assuming that their combined effects were solely additive. Comparisons were conducted using 2 Gy single-dose IR to maintain clinical relevance. AZD0156 led to lower-than-expected surviving fractions in both cell lines, although statistical significance was only achieved in MDA-MB-231 cells (*p* = 0.0146, Figure 5C,D).

### 3.4. AZD0156, AZD6738 and Olaparib Sensitize MDA-MB-231 and MCF-7 Cells to 24-Hour Fractionated IR

Combinations of 24-h fractionated IR and DDR inhibitors were then investigated. MDA-MB-231 and MCF-7 cells were treated with AZD0156 (0.01 µM), AZD6738 (0.25 µM) or olaparib (1 µM) one hour prior to the initial irradiation, and the second IR dose was delivered 24 h later. Total IR doses of 1 to 8 Gy were administered, and post-treatment cell survival was quantified with clonogenic assays. The most profound radiosensitization was again observed with the use of AZD0156 in MDA-MB-231 cells, as no colonies formed at a total IR dose of 4 Gy. However, unlike for single-dose exposures, both olaparib and AZD6738 also showed significant radiosensitization when fractionated exposures were delivered (Figure 6A). Similarly, for MCF-7 cells, all three DDR inhibitors increased sensitivity to 24-h fractionated IR where no effects were seen for single doses (Figure 6B).

The impact of inhibitors on split-dose recovery was also investigated by comparing single-dose and fractionated IR for different inhibitor treatments. When treated with any of the DDR inhibitors, split-dose recovery was almost completely abrogated in MDA-MB-231 cells compared to IR-only exposures. Similarly, while survival increased slightly following fractionation for MCF-7 cells, this was significantly diminished compared to IR-only experiments (Figure 7).

To quantitatively summarize these data, the mean inactivation dose ($\bar{D}$) and SER of each treatment group were calculated and presented in Table 2. For single-dose IR, only MDA-MB-231 cells treated with AZD0156 showed a significant change in sensitivity. In contrast, for 24-h fractionated IR, all DDR inhibitors significantly sensitized both cell lines to IR. Some cell-line specificity is also seen, with AZD0156 being dramatically more potent in MDA-MB-231 cells, while AZD0156 and AZD6738 were similarly effective in MCF-7 cells.

### 3.5. DDR Inhibition Influences SLD Repair in MDA-MB-231 and MCF-7 Cells

To determine if DDR inhibition affects SLD repair, MDA-MB-231 and MCF-7 cells were first treated with AZD0156 (0.01 µM), AZD6738 (0.25 µM) or olaparib (1 µM). They were irradiated an hour later with two IR doses separated by 0 to 4 h. Doses were selected to give similar levels of survival to two fractions of 3 Gy IR alone (Figure 4). A total IR dose of 3 Gy was selected for all treatment groups other than MDA-MB-231 cells treated with AZD0156, whose greater sensitivity required a lower total dose of 2 Gy. Clonogenic assays were performed to assess cell survival, and the surviving fractions were graphed using the one-phase exponential model. As the interval between IR doses increased, survival of MDA-MB-231 cells treated with AZD0156 or olaparib showed a slight increase before plateauing at approximately the 2-h timepoint; cells treated with AZD6738, however, did not show changes in survival with time (Figure 8A). On the other hand, MCF-7 cells demonstrated increasing survival regardless of the DDR inhibitors used, although this trend was less noticeable after treatment with AZD6738 (Figure 8B).

Next, SLD repair half-times (*t*_1⁄2_) and ratios of the surviving fraction after 4-h fractionated IR to non-fractionated IR (SF_4h_:SF_0h_) were listed in Table 3 to enable quantitative analysis of SLD repair with and without DDR inhibition. As no significant repair was seen for either cell line when treated with AZD6738, the *t*_1⁄2_ values could not be satisfactorily fitted. Compared to irradiation only, no clear pattern is seen in the rate of SLD with AZD0156 or olaparib. MDA-MB-231 cells experienced a decrease of about 10 min in SLD repair half-times after treatment. In contrast, in MCF-7 cells, the same treatments led to longer repair half-times, with the largest increase of approximately 16 min caused by olaparib. However, none of these changes were statistically significant, and these repair half-times are still significantly shorter than the 24-h gap where significant impacts on sensitivity are seen with drug treatment above.

With regards to the SF_4h_:SF_0h_ values, they were decreased by all DDR inhibitors, especially AZD6738, in MDA-MB-231 cells. Cell survival was ~3 times higher after 4 h than at time 0 for the MDA-MB-231 IR-only group, compared with ~2, ~1.2 and ~1.1 times with AZD0156, olaparib and AZD6738 (Table 3). In contrast, there was no consistent trend in MCF-7 cells when compared with the IR-only group, as AZD0156 and olaparib led to increases in recovery, whereas the opposite was found with AZD6738. It must be noted that the IR-only treatment groups were irradiated with a higher IR dose than the IR and DDR inhibitor combination treatment groups, the implications of which will be further discussed below.

### 3.6. AZD0156, AZD6738 and Olaparib Delay the Repair of IR-Induced DNA DSBs

Immunofluorescence experiments were conducted to evaluate the influence of AZD0156 (0.01 µM), AZD6738 (0.25 µM) and olaparib (1 µM) on the formation and resolution of DNA damage, using 53BP1 foci as markers of DNA DSBs. When cells were irradiated with 2 Gy, all treatment groups except for MDA-MB-231 cells treated with IR and olaparib demonstrated most foci 30-min post-IR, gradually decreasing over time. Notably, all DDR inhibitors delayed foci resolution in both cell lines, subsequently leading to more residual foci compared to IR-only treatment at 24-h post-IR. In MDA-MB-231 cells, this retardation was most prominent with AZD0156, whereas in MCF-7 cells, it was comparable across the three drugs (Figure 9).

### 3.7. Effect of AZD0156, AZD6738 and Olaparib on Activity of Target Proteins

To evaluate the effect of AZD0156, AZD6738 and olaparib on their respective DNA repair pathways, phosphorylation of ATM, phosphorylation of ATR and protein level of PAR were evaluated in MDA-MB-231 and MCF-7 cells treated with 4 Gy IR, a DDR inhibitor or combinations of both (Figure 10 and Appendix A). Treatment with 1 µM of olaparib effectively reduced levels of PAR in both cell lines, particularly after irradiation. Cells treated with 0.25 µM of AZD6738, an ATR inhibitor, saw a significant reduction in phosphorylation of ATR in both cell lines following irradiation. While this was more pronounced in the MCF-7 cells, this had only a small impact on levels of cell death after irradiation. Cells treated with 0.01 µM AZD0156, an ATM inhibitor, saw reduced phosphorylation of ATM following irradiation. Interestingly, this reduction in phosphorylation is more pronounced in MCF-7 than in MDA-MB-231 cells, seemingly in contrast to the observed IC_50_ values (MDA-MB-231 0.011 µM, MCF-7 0.25 µM). This may indicate a greater dependence of MDA-MB-231 cells on ATM or potential off-target effects in these cells.

### 3.8. Effect of AZD0156, AZD6738 and Olaparib on the Cell Cycle Distribution of MDA-MB-231 and MCF-7 Cells

To evaluate the cell cycle distribution of MDA-MB-231 and MCF-7 cells treated with 2 Gy IR, a DDR inhibitor or combinations of both, cell cycle analysis was conducted using flow cytometry at 1 and 24 h after the indicated treatments. After an hour, no changes were observed across all treatment groups in both cell lines (Figure 11A,B). At 24 h in MDA-MB-231 cells, 2 Gy IR increased G_2_/M arrest slightly, whereas treatment with only AZD6738 or olaparib influenced G_1_ arrest, leading to an increase and decrease, respectively (Figure 11C). As for MCF-7 cells, there were no alterations whatsoever akin to the previous time point (Figure 11D).

## 4. Discussion

### 4.1. Effect of AZD0156, AZD6738 and Olaparib on Single-Dose and Fractionated IR Sensitivity

To our knowledge, this study is the first to directly compare the effects of DDR inhibition on single-dose IR against fractionated IR in vitro using cell survival as the response variable. In single-dose IR experiments, only AZD0156 induced significant radiosensitization. On the other hand, radiosensitization by all DDR inhibitors was observed when combined with 24-h fractionated IR. The vast difference between single-dose and fractionated IR experiments reflects previous studies on the radiosensitizing effect of olaparib, which generated varying conclusions depending on whether single-dose or fractionated IR was utilized [21,26]. These results underline the need to incorporate IR dose fractionation into preclinical studies to mimic clinical radiotherapy regimens and their interaction with DDR inhibition. Doing so will improve the clinical translation of future preclinical studies involving these combination therapies.

Another important discussion point is the influence of DDR inhibition on the survival benefit of IR dose fractionation. In IR-only experiments, significant recovery is expected for 24-h fractionated compared to single-dose IR. However, treatment with any of the three DDR inhibitors significantly diminished the increase in survival after fractionation in both cell lines. These findings suggest that DDR inhibitors enhance radiosensitivity following fractionated treatment by disrupting recovery between fractionated IR doses to a greater extent than would be expected from single-dose observations.

When considering fractionated exposure with DDR inhibitors, it is plausible that disruption of the DDR effectively delays the complete resolution of IR-induced DNA damage for up to 24 h, thus generating a cellular environment with heightened radiosensitivity for delivery of a second fraction. This is further supported by data from the immunofluorescence experiments, which indicated that all inhibitors increased the number of residual DNA DSBs at 24 h after 2 Gy IR. However, significant DNA repair is still seen, highlighting the possible role of other mechanisms in fractionated sensitivity. It is vital to explore the mechanisms by which these drugs increase radiosensitivity.

Among the three target proteins in this study, ATM is the only one that is traditionally assumed to play a major role in repairing DNA DSBs, which are the most lethal form of DNA damage [5]. Therefore, it is unsurprising that the ATM inhibitor AZD0156 was the strongest radiosensitizer. AZD0156 prevented phosphorylation of ATM, a key component of the DDR system, and so may play a crucial role in delaying DNA damage repair and increasing cell radiosensitization between fractions. Nonetheless, the other two drugs also showed radiosensitizing effects, albeit at smaller levels. Cells treated with AZD6738 will be less able to repair single-stranded DNA intermediates via ATR. Subsequently, these lesions may be transformed into DSBs at replication forks as cells continue to synthesize DNA after the initial IR dose [27]. Coupled with a second IR dose 24 h later, these lesions may be aggravated and consequently cause cellular death. As for olaparib, its radiosensitizing ability can be attributed to the involvement of PARP in both SSB and DSB repair [10]. Yet, the degree of radiosensitization observed was lower than that by AZD0156, therefore implying that ATM is more important to effective DSB repair than PARP. This is consistent with a study demonstrating that irradiated cells tend to activate non-PARP-mediated processes to conduct DSB repair [28]. The different mechanisms of these three DDR inhibitors may explain their varying radiosensitizing capabilities. Differences in repair capacity of these cell lines due to variations in other DNA repair genes have been explored in detail elsewhere [29,30].

### 4.2. Effect of AZD0156, AZD6738 and Olaparib on SLD Repair

As SLD repair contributes to the increase in cell survival after IR dose fractionation, our study sought to investigate if DDR inhibition induces radiosensitization by impairing such repair. In IR-only treatment groups, a time-dependent increase in survival fraction was clearly observed. This data is in excellent agreement with previously published data in a study involving Chinese Hamster V79 cells, where similar SLD repair times and increased survival was reported [31].

In irradiated MDA-MB-231 cells, all three DDR inhibitors reduced the SLD repair capacity. Counter-intuitively, AZD0156 and olaparib decreased the SLD repair half-time, which usually implies that SLD repair was carried out more rapidly, thus favoring greater cellular survival. This was clearly contradictory to the drugs’ observed enhancement of 24-h fractionated IR. To explain this discrepancy, it is possible that the reduced amount of SLD repair possible after DDR inhibition can be completed in a shorter duration if, for example, more complex breaks fail to repair. Therefore, these changes in SLD suggest that DDR inhibitors radiosensitize MDA-MB-231 cells by reducing overall SLD repair capacity but not the rate of repair. This can be directly related to lower levels of activation of ATM, ATR and PARP, proteins that will mediate DNA damage response. The impact of the DDR inhibitors on SLD repair in MCF-7 cells was more varied. AZD0156 and olaparib showed both increased SLD repair half-time and repair capacity, indicating a complex interplay between fractionated IR and DDR inhibition.

When interpreting these results, it is important to note the differences in the IR doses utilized. Cells treated with IR only were exposed to a dose of 6 Gy, whereas half of this was used for cells treated with DDR inhibitors (and only 2 Gy for MDA-MB-231 cells treated with AZD0156). This was done to enable appreciable colony formation in cells after DDR inhibition. In many conditions, the overall survival level is similar (particularly for MCF-7 cells). This indicates significant sensitization by these inhibitors in these fractionated combinations, independent of the observed SLD. This suggests DDR inhibition gives rise to classes of damage that fail to repair and are available to interact with subsequent fractions for long periods beyond those considered in this work.

This does add some complexity in interpreting SLD differences, however. This analysis made the assumption that SLD repair kinetics and capacity are independent of IR intensity, which is backed by a study demonstrating similar repair half-times across different IR fraction sizes [32]. Yet, it cannot be ruled out that higher IR doses generate more extensive and complicated DNA lesions, thus influencing repair dynamics in an unforeseeable manner. More detailed explorations of the interplay between IR dose and SLD may improve validity and yield more robust findings.

Finally, differences are seen in the impact of DDR inhibition between MCF-7 and MDA-MB-231 cells, even though both cell lines were similarly sensitized to fractionated IR. This may be explained by the extensive nature of the DDR. Even if one DDR pathway was impeded, alternative processes could be activated to mediate SLD repair after irradiation. Moreover, proficiency of the compensatory mechanisms may vary among cell lines and the DDR inhibitors used, thus explaining the diverse findings in different cancer cell lines. Altogether, while SLD repair studies showed some impact of DDR inhibition, they cannot explain the extent of radiosensitization.

### 4.3. Cell Cycle Modifications Do Not Contribute to Radiosensitization by AZD0156, AZD6738 and Olaparib

Other than conducting DNA repair, another important function of the DDR is regulating cell cycle checkpoints. As cells in different cell cycle phases exhibit varying susceptibility to IR [33], we investigated whether altering cell cycle distribution is a mechanism by which DDR inhibition enhances radiosensitization. Since the S phase is known to be particularly radioresistant [34], the roles of ATM and ATR in mediating S-phase arrest designate their inhibitors AZD0156 and AZD6738 as noteworthy subjects in cell cycle analysis. The key time points considered were 1 h after drug treatment and 24 h after IR with drug treatment, as they represent points of IR exposure according to our experimental design. However, no significant changes in cell cycle distribution were observed under these circumstances, indicating that cell cycle changes do not mediate radiosensitization by DDR inhibition.

### 4.4. Clinical Implications

The most important clinical implication that can be inferred from this study is the exemplary radiosensitizing potential of AZD0156. Based on our data, it was the only drug that potentiated single-dose IR. More importantly, it was the strongest enhancer of fractionated IR in MDA-MB-231 and MCF-7 cells. The observed radiosensitizing effects were achieved at a relatively low drug concentration of 0.01 µM, thus indicating a therapeutic window for this drug since a less aggressive dosage may be sufficient for radiosensitization. Furthermore, the significantly higher activity of AZD0156 in MDA-MB-231 cells implies that certain cancer phenotypes may be more responsive to this DDR inhibitor. If the higher ATM expression in MDA-MB-231 than MCF-7 cells truly signifies greater reliance on ATM-mediated DNA repair pathways [35], then utilizing ATM as a predictive biomarker for AZD0156 and other ATM inhibitors may be a prospect worth exploring in future research. Our study shows a clear difference in AZD0156 IC_50_ between cell lines. Consequently, the capability to activate important proteins of the targeted DDR pathway may also be distinct between cell lines. Altogether, our study provides substantial proof that AZD0156 is an effective radiosensitizer, thus encouraging its further evaluation in clinical trials.

Other than AZD0156, AZD6738 and olaparib were also assessed in this preclinical study. Although their radiosensitizing capability was not as potent as that of the ATM inhibitor, our findings still justify their utilization in combination with IR to boost cancer killing. More reassuring was that these DDR inhibitors potentiated fractionated IR, further strengthening their clinical relevance given the predominance of fractionated radiotherapy in cancer management nowadays. Additionally, the results serve to rationalize the ongoing clinical trials evaluating AZD6738 (NCT02223923) and olaparib (NCT03109080) in combination with radiotherapy. All things considered, ATR and PARP should be considered viable therapeutic targets for radiosensitizing strategies, even though they may not be as important as ATM with regards to their roles in DNA DSB repair.

### 4.5. Limitations

A number of limitations in this study must be considered while interpreting the findings. First, all experiments were conducted in vitro utilizing breast cancer cell lines. Although useful for initial exploration of the interaction between IR and DDR inhibition, this cell-only approach does not fully reflect true cancer dynamics and microenvironment. To address this, further in vivo studies should be performed to account for the tumor microenvironment and its influence on radiosensitivity [36]. Second, the methods by which drug treatments were delivered in this study may impact the validity of the results. In experiments involving DDR inhibition, cells were constantly exposed to drugged media until they were assayed. Although this ensured that the targeted DDR enzymes were sufficiently suppressed, it was not representative of in vivo pharmacokinetic properties.

More advanced preclinical studies in this area are needed to refine further our understanding of the synergy between IR and various forms of DDR inhibition. Two particular areas of focus should be (1) drug exposure times before irradiation to inform the optimization of treatment schedules, especially if cells are found to be exceptionally radiosensitive after a certain period of DDR inhibition; (2) the effects of combining IR and DDR inhibition on non-malignant cells, to determine potential off-target treatment toxicity.

Data presented here highlight the importance of identifying biomarkers for combination treatments with DDR inhibitors. The radiosensitizing effect of AZD0156 was more prominent in MDA-MB-231 than MCF-7 cells, which may be attributed to the difference in ATM expression between the cell lines [35]. Validation of this association through further experiments will be required to establish a causal relationship. In addition, assessment of the correlation between ATR or PARP expression and the efficacy of AZD6738 or olaparib, respectively, could further inform the utility of target protein levels as predictive biomarkers of DDR inhibitors. Moreover, including other DDR effectors and pharmacological inhibitors in biomarker studies, as well as concurrent characterization of mutations in relevant genes such as *BRCA* and *p53*, could determine if defects in certain DDR pathways confer better response to specific drugs. Altogether, the therapeutic window of DDR inhibitors will certainly be widened by the discovery of robust predictive biomarkers, as demonstrated by the synthetic lethality of PARP inhibitors and *BRCA* loss [37].

## 5. Conclusions

DDR inhibitors have demonstrated preclinical radiosensitization, yet translation into the clinic has been disappointing. As the main type of radiotherapy used clinically is fractionated radiotherapy, a lack of preclinical studies evaluating DDR inhibitors’ influence on single-dose and fractionated radiotherapy regimens could explain the suboptimal results. To address this gap in the literature, our study investigated the radiosensitizing effect of the ATM inhibitor AZD0156, ATR inhibitor AZD6738, and PARP inhibitor olaparib on human breast cancer cells, utilizing single-dose and fractionated radiotherapy. Our study has shown that AZD0156, AZD6738 and olaparib can induce radiosensitization in preclinical breast cancer models. More importantly, it was demonstrated that DDR inhibitors impact SLD repair capability and induce higher levels of persistent damage after irradiation, having a bigger impact in fractionated radiotherapy than in single-dose radiotherapy. This implies that preclinical studies involving similar drugs should employ single-dose and fractionated radiotherapy regimens to assess radiosensitizing potential accurately. Further studies on the underlying mechanisms by which DDR inhibition induces radiosensitization and the potential side effects of combining IR with DDR inhibitors will be crucial for optimizing these combination treatments.

## Figures and Tables

**Figure 1 cancers-14-03794-f001:**
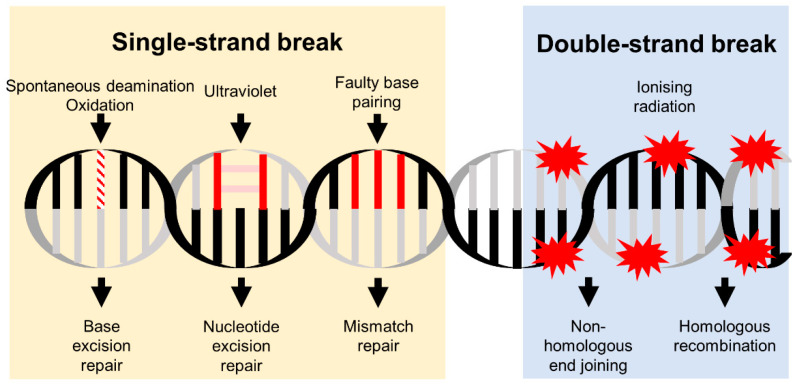
Main causes and repair mechanisms of DNA single-strand and double-strand breaks. Various endogenous or exogenous factors can damage DNA. Consequently, the DDR utilizes multiple pathways to repair different forms of DNA lesions.

**Figure 2 cancers-14-03794-f002:**
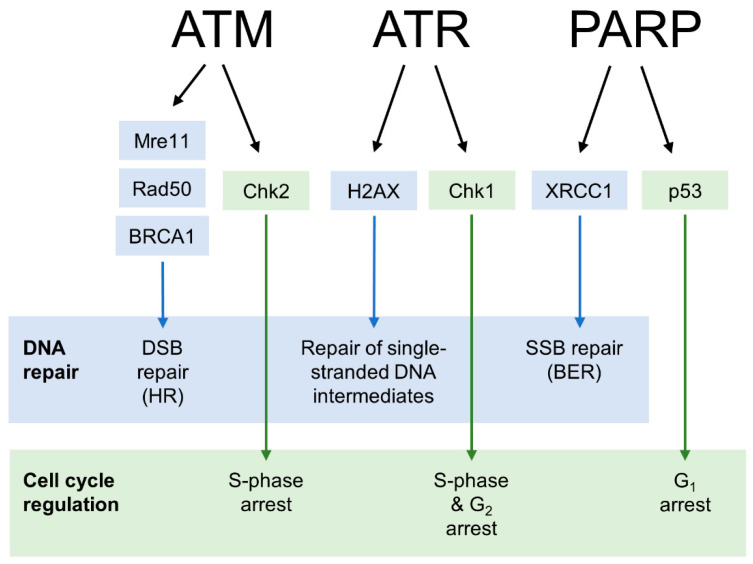
Main substrates of ATM, ATR, and PARP in the DDR. Activation of downstream proteins by the three signal transducers is required for various DNA repair and cell cycle regulation pathways.

**Figure 3 cancers-14-03794-f003:**
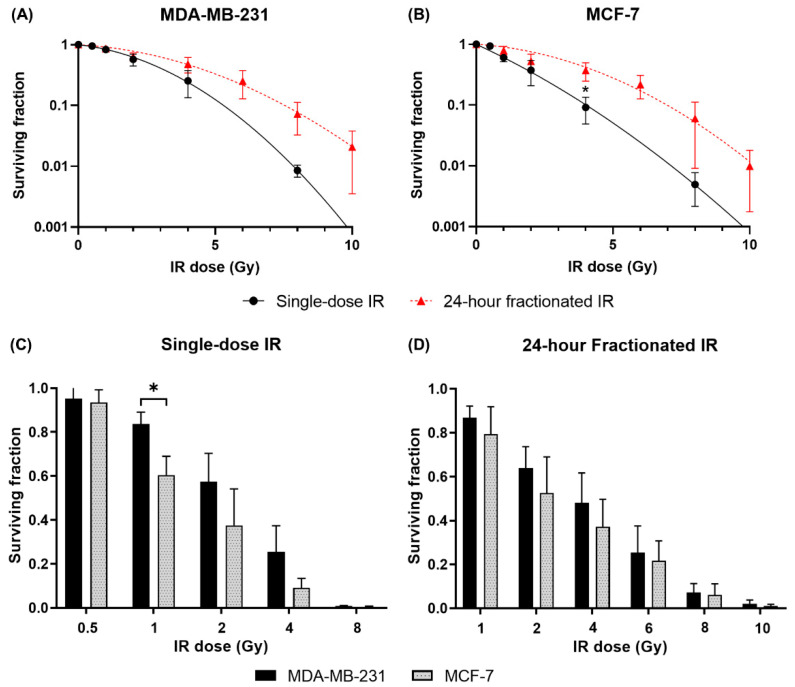
IR dose fractionation leads to decreased lethality, and MCF-7 cells are more radiosensitive than MDA-MB-231 cells. Clonogenic assays were performed to quantify cell survival after treatment with either a single-dose IR or 24-h fractionated IR at total doses of 0.5 to 10 Gy. Surviving fractions of (**A**) MDA-MB-231 and (**B**) MCF-7 cells were fitted to graphs using the linear-quadratic model. The effects of (**C**) single-dose and (**D**) 24-h fractionated IR on the cell lines were then compared. Data are shown as mean ± SD of three independent experiments performed in duplicate or triplicate wells. * *p* < 0.05 difference between single-dose and fractionated survival.

**Figure 4 cancers-14-03794-f004:**
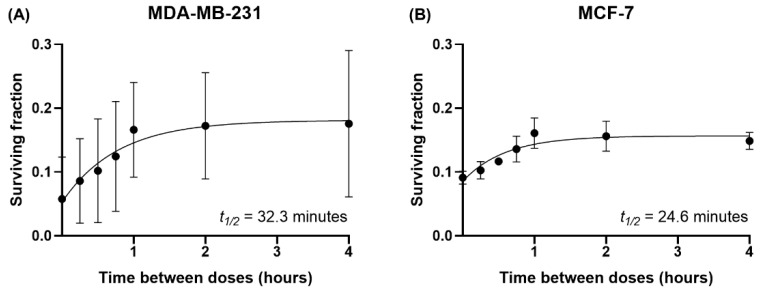
MDA-MB-231 and MCF-7 cells demonstrate similar SLD repair kinetics. Clonogenic assays were used to quantify cell survival after treatment with two 3 Gy IR doses separated by 0–4 h. Surviving fractions of (**A**) MDA-MB-231 and (**B**) MCF-7 cells were fitted to graphs using the one-phase exponential model, and the estimated SLD repair half-time (*t*_1/2_) values were obtained. Data are shown as mean ± SD of three independent experiments performed in triplicate wells.

**Figure 5 cancers-14-03794-f005:**
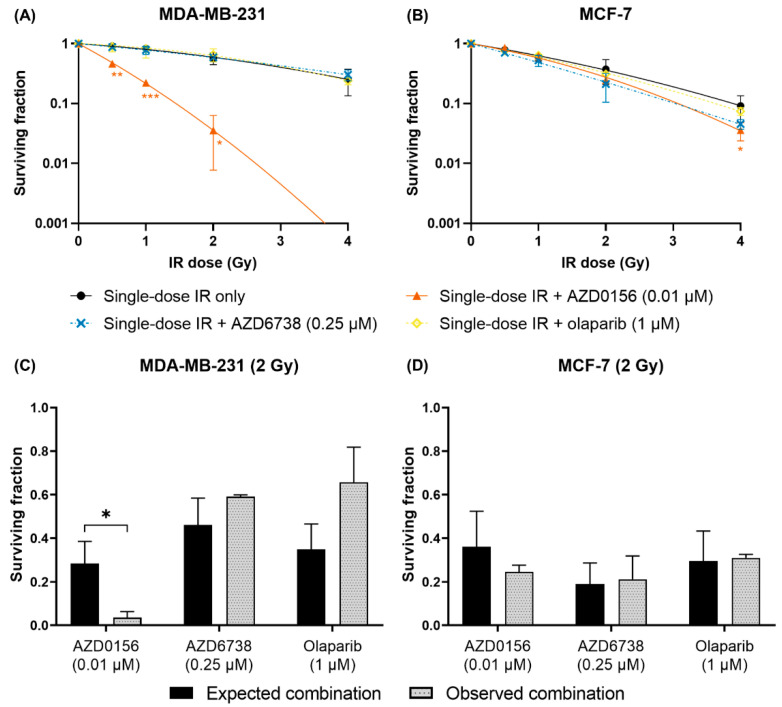
AZD0156 potentiates single-dose IR in MDA-MB-231 cells. (**A**) MDA-MB-231 and (**B**) MCF-7 cells were treated with AZD0156 (0.01 µM), AZD6738 (0.25 µM) or olaparib (1 µM) one hour prior to single-dose irradiation at 0.5 to 4 Gy, then cell survival was quantified with clonogenic assays. Survival curves of cells treated with single-dose IR only from previous experiments were displayed for comparison. To determine potential synergy between IR and DDR inhibition, surviving fractions of drugged (**C**) MDA-MB-231 and (**D**) MCF-7 cells irradiated at 2 Gy were compared. The expected surviving fractions from combination treatments were calculated as products of the observed surviving fractions after treatment with single-dose IR only and DDR inhibitor only. Data are shown as mean ± SD of three independent experiments performed in duplicate or triplicate wells. In (**A**,**B**), * *p* < 0.05, ** *p* < 0.01, *** *p* < 0.001 vs. single-dose IR only. In (**C**), * *p* < 0.05.

**Figure 6 cancers-14-03794-f006:**
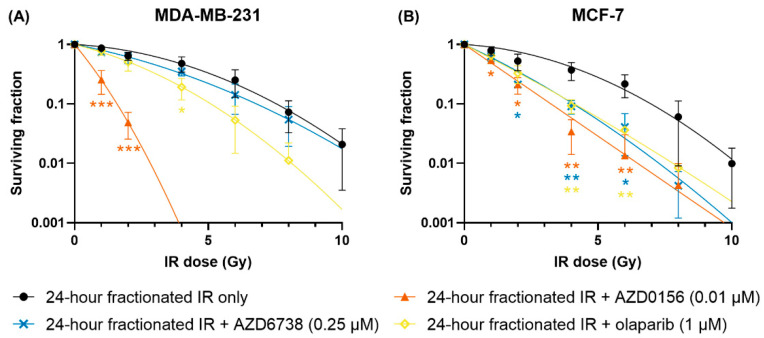
IR dose fractionation enhances the radiosensitizing effect of DDR inhibitors. (**A**) MDA-MB-231 and (**B**) MCF-7 cells were treated with AZD0156 (0.01 µM), AZD6738 (0.25 µM) or olaparib (1 µM) one hour prior to 24-h fractionated irradiation at 1 to 8 Gy, then cell survival was quantified with clonogenic assays. Survival curves of cells treated with 24-h fractionated IR only from previous experiments were displayed for comparison. Data are shown as mean ± SD of three independent experiments performed in duplicate or triplicate wells. * *p* < 0.05, ** *p* < 0.01, *** *p* < 0.001 vs. 24-h fractionated IR only.

**Figure 7 cancers-14-03794-f007:**
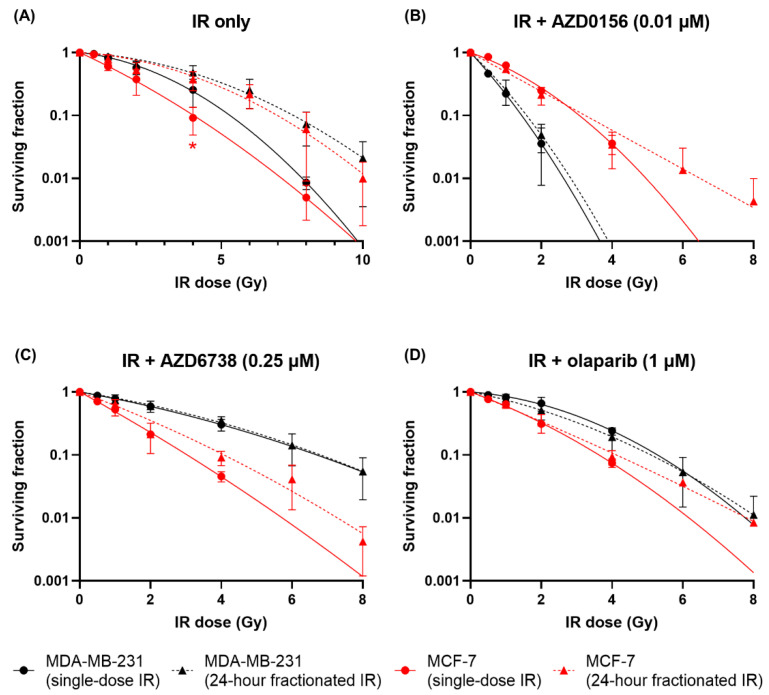
DDR inhibitors reduce the survival benefit of IR dose fractionation. The survival curves of MDA-MB-231 and MCF-7 cells treated with single-dose or 24-h fractionated IR with or without DDR inhibition were compared. Data are shown as mean ± SD of three independent experiments performed in duplicate or triplicate wells. * *p* < 0.05 for single-dose IR vs. 24-h fractionated IR.

**Figure 8 cancers-14-03794-f008:**
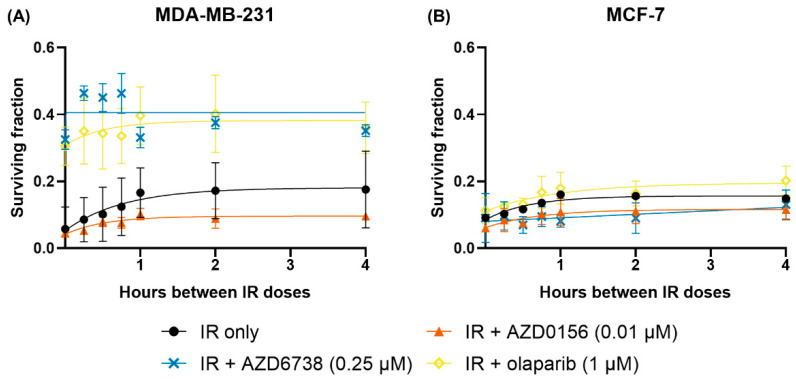
AZD6738 attenuates SLD repair in MDA-MB-231 cells. (**A**) MDA-MB-231 and (**B**) MCF-7 cells were treated with AZD0156 (0.01 µM), AZD6738 (0.25 µM) or olaparib (1 µM) one hour prior to irradiation with two 1.5 Gy doses separated by 0 to 4 h, except for MDA-MB-231 cells treated with AZD0156 in which two 1 Gy doses were used. Cell survival was then quantified with clonogenic assays, and the surviving fractions were fitted to graphs using the one-phase exponential model. The survival curves of cells treated with two 3 Gy IR doses were only displayed for comparison. Data are shown as mean ± SD of three independent experiments performed in duplicate wells.

**Figure 9 cancers-14-03794-f009:**
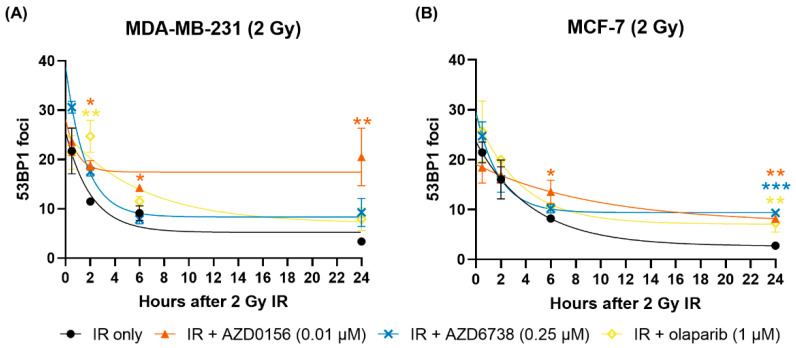
DDR inhibitors delay the resolution of 53BP1 foci in irradiated cells. Immunofluorescence experiments were conducted to assess DNA DSB repair kinetics in irradiated (**A**) MDA-MB-231 and (**B**) MCF-7 cells, using 53BP1 foci as DSB markers. Cells were drugged one hour before irradiation. Then they were incubated for the indicated durations after 2 Gy IR before being assayed. The average number of 53BP1 foci in 50 cells from each sample was calculated, and results from the IR experiments were fitted to graphs using the one-phase decay exponential model. Data are shown as mean ± SD of at least two independent experiments. * *p* < 0.05, ** *p* < 0.01, *** *p* < 0.001 vs. IR only.

**Figure 10 cancers-14-03794-f010:**
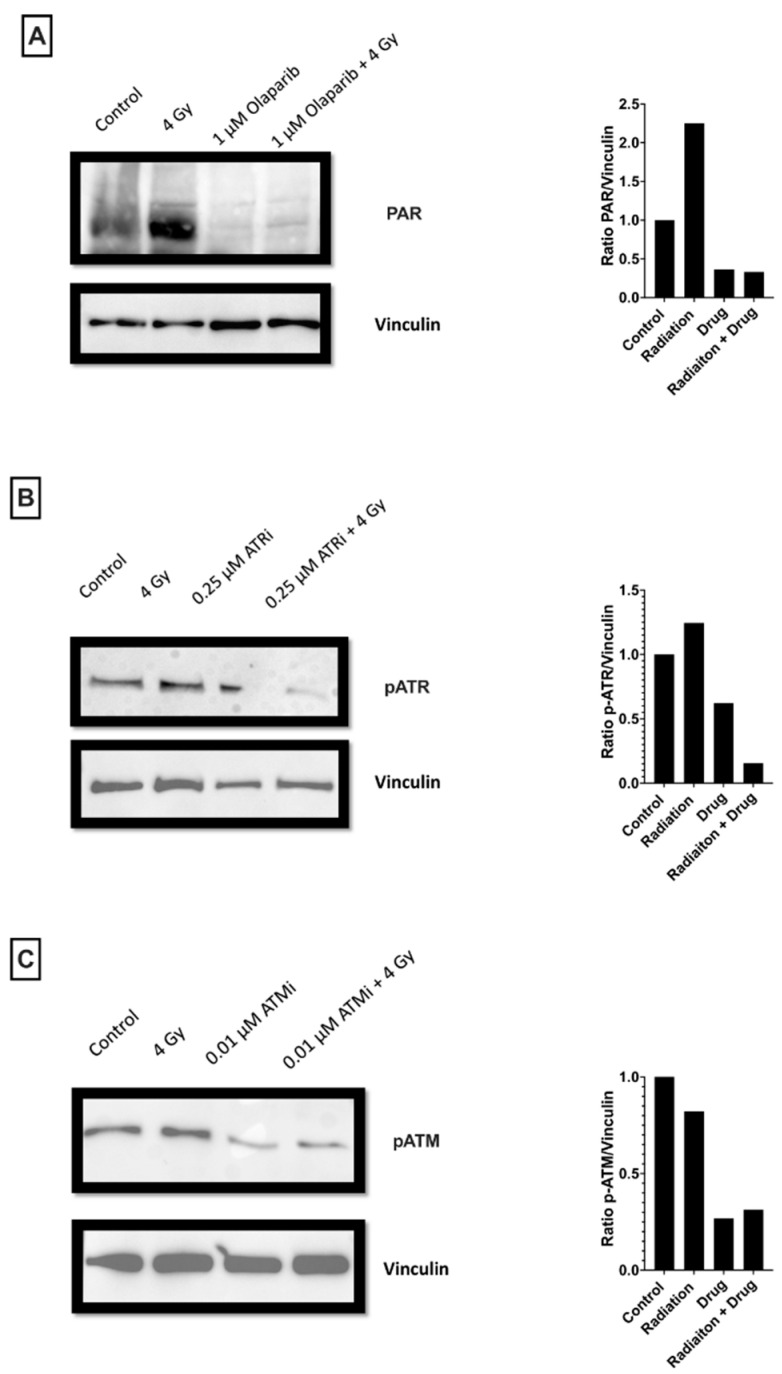
Effect of (**A**) olaparib, (**B**) AZD6738 and (**C**) AZD0156 on target proteins of respective DNA repair pathways in MCF-7 cells. Olaparib directly affects the protein levels of PAR. AZD6738, an ATR inhibitor, directly affects the phosphorylation of ATR. AZD0156, an ATM inhibitor, directly affects the phosphorylation of ATM. Phosphorylation or protein levels were evaluated for control cell populations only treated with DMSO, cells only treated with 4 Gy of radiation, cells only treated with DDR inhibitors, or cells treated with a combination of both. Levels of Vinculin were used as a loading control, and quantification was performed with ImageJ. Protein ratios were normalized to the untreated control cells.

**Figure 11 cancers-14-03794-f011:**
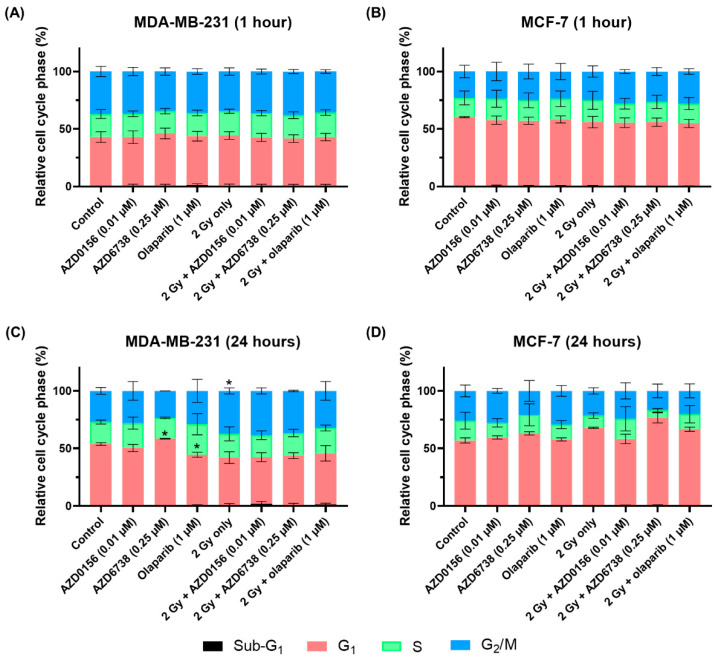
DDR inhibitors have minimal effect on cell cycle distribution of MDA-MB-231 and MCF-7 cells. MDA-MB-231 and MCF-7 cells were treated with 2 Gy IR, a DDR inhibitor, or combinations of both, then cell cycle analysis was performed after (**A**,**B**) 1 and (**C**,**D**) 24 h using flow cytometry. Data shown as mean ± SD of at least two independent experiments, with at least 5000 cells analyzed in each sample. * *p* < 0.05 vs. control.

**Table 1 cancers-14-03794-t001:** IC_50_ (µM) of AZD0156, AZD6738, and olaparib in MDA-MB-231 and MCF-7 cells as determined by clonogenic assays. IC_50_ values were estimated from the plotted dose-response curves (not shown). Data are shown as mean ± SD of three independent experiments performed in duplicate wells.

DDR Inhibitor	MDA-MB-231 IC_50_ (μM)	MCF-7 IC_50_ (μM)
AZD0156	0.011 ± 0.005	0.25 ± 0.07
AZD6738	0.30 ± 0.10	0.18 ± 0.06
Olaparib	1.3 ± 0.6	1.5 ± 0.8

**Table 2 cancers-14-03794-t002:** Mean inactivation dose ($\bar{D}$ ) of single-dose and 24-h fractionated IR with and without DDR inhibition and the sensitizer enhancement ratio (SER) of each DDR inhibitor in MDA-MB-231 and MCF-7 cells. $\bar{D}$ values were estimated by calculating the area under the survival curves in Figure 5 and Figure 6. SER values were calculated as the $\bar{D}$ for the IR-only groups divided by the $\bar{D}$ for the IR and drug combination groups. Data are shown as mean ± SD of three independent experiments performed in duplicate or triplicate wells.

Treatment	MDA-MB-231		MCF-7	
	$\bar{D}$ (Gy)	SER	$\bar{D}$ (Gy)	SER
**Single-dose IR**				
IR only	2.5 ± 0.2		1.8 ± 0.2	
IR + AZD0156 (0.01 µM)	0.67 ± 0.02	3.7 ± 0.3	1.55 ± 0.06	1.2 ± 0.1
IR + AZD6738 (0.25 µM)	2.46 ± 0.09	1.02 ± 0.09	1.4 ± 0.1	1.3 ± 0.2
IR + olaparib (1 µM)	2.5 ± 0.2	1.0 ± 0.1	1.65 ± 0.06	1.1 ± 0.1
**24-h fractionated IR**				
IR only	4.0 ± 0.3		3.4 ± 0.3	
IR + AZD0156 (0.01 µM)	0.78 ± 0.08	5.1 ± 0.7	1.46 ± 0.09	2.3 ± 0.3
IR + AZD6738 (0.25 µM)	3.2 ± 0.2	1.3 ± 0.1	1.7 ± 0.1	2 ± 0.2
IR + olaparib (1 µM)	2.6 ± 0.2	1.5 ± 0.2	1.9 ± 0.1	1.8 ± 0.2

**Table 3 cancers-14-03794-t003:** SLD repair half-time (*t*_1⁄2_) in minutes and ratio of the surviving fraction after 4-h fractionated IR to non-fractionated IR (SF_4h_:SF_0h_) of MDA-MB-231 and MCF-7 cells treated with IR only and in combination with AZD0156 (0.01 µM), AZD6738 (0.25 µM) or olaparib (1 µM). In the AZD0156 treatment group, MDA-MB-231 cells were irradiated with two 1 Gy doses, while MCF-7 cells were irradiated with two 1.5 Gy doses. *t*_1⁄2_ values were estimated from the survival curves in Figure 8, and values for the two 1.5 Gy IR + AZD6738 (0.25 µM) groups were unavailable due to ambiguous or interrupted fits with the one-phase exponential model. Data were obtained from three independent experiments performed in duplicate or triplicate wells.

Treatment	MDA-MB-231		MCF-7	
	*t*_1⁄2_ (Minutes)	SF_4h_:SF_0h_	*t*_1⁄2_ (Minutes)	SF_4h_:SF_0h_
2 × 3 Gy IR only	32.3	3.05	24.6	1.63
2 × 1/1.5 Gy IR + AZD0156 (0.01 µM)	22.0	2.11	28.9	1.90
2 × 1.5 Gy IR + AZD6738 (0.25 µM)	-	1.08	-	1.31
2 × 1.5 Gy IR + olaparib (1 µM)	21.9	1.18	41.0	1.78

## Data Availability

The data used to generate all figures in this paper are made available in the Appendix A.

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
