# Peer review of "DNA Repair Inhibitors Potentiate Fractionated Radiotherapy More Than Single-Dose Radiotherapy in Breast Cancer Cells"

_cancers, 2022, doi:10.3390/cancers14153794_

Round 1

Reviewer 1 Report

This is a revised submission analysing the impact of inhibitors targeting ATM, ATR and PARP on the radiosensitisation of two breast cancer cell lines using both single dose and fractionated x-ray radiation. The authors have provided some additional data and appropriately addressed some of my comments suggested on the previous version.

However, I still have two concerns. The first is that the new data (Figure 11) isn’t very supportive that the inhibitors are functionally active in both cell lines. Whilst it’s convincing that AZD0156 supresses CHK2 phosphorylation in MDA-MB-231 cells (Figure 11A, left panel) and AZD6738 supresses CHK1 activity in MCF-7 cells (Figure 11B, right panel), the other data are very weak. In some cases the levels of the marker protein are higher in the unirradiated controls than the 4 Gy irradiated samples (e.g. Figure 11A, right panel; Figure 11C, left panel). To address this, the authors should either replace with higher quality images reflective of kinase inhibition, or in the case of AZD0156 and AZD6738, possibly analyse direct ATM and ATR phosphorylation/activation themselves. I would also suggest quantifying the blots and providing the relative degree of inhibition (either graphically, or numerically below the figures).

The second comment highlighted from the previous review is about the lack of mechanistic evidence supporting the differential response of the two cell lines to the combination of the inhibitors plus single and fractionated radiation doses. I agree with the authors that this would require additional experimentation, although this is not completely beyond scope. Based on the results provided, it is therefore difficult to predict based on just the two cell lines utilised generating different responses, how broadly applicable and effective the strategy of DDR inhibition with fractionated radiation in in vitro breast cancer models is and what are the key driving factors for this response. In this respect, the paper is limited, although this does provides some interesting observations.

Author Response

We would like to thank the reviewer for their positive feedback on the manuscript, and their further useful suggestions. We would also like to apologise for the delay in responding to these requests, as radiation work in our lab was temporarily delayed due to equipment failures. These have now been resolved, and we have revised the manuscript with respect to these suggestions, as described below.

Point 1:

The first is that the new data (Figure 11) isn’t very supportive that the inhibitors are functionally active in both cell lines. Whilst it’s convincing that AZD0156 supresses CHK2 phosphorylation in MDA-MB-231 cells (Figure 11A, left panel) and AZD6738 supresses CHK1 activity in MCF-7 cells (Figure 11B, right panel), the other data are very weak. In some cases the levels of the marker protein are higher in the unirradiated controls than the 4 Gy irradiated samples (e.g. Figure 11A, right panel; Figure 11C, left panel). To address this, the authors should either replace with higher quality images reflective of kinase inhibition, or in the case of AZD0156 and AZD6738, possibly analyse direct ATM and ATR phosphorylation/activation themselves. I would also suggest quantifying the blots and providing the relative degree of inhibition (either graphically, or numerically below the figures).

In response to the reviewer’s suggestion, we have performed Western blots using pATM, pATR, and PAR as target proteins of AZD0156, AZD6738 and olaparib, respectively. Results are shown in the new Figure 10, a new supplementary figure, and further described in Section 3.7. The blots were also quantified to show the relative degree of inhibition graphically.

Changes in expression or phosphorylation of key targets are in agreement with the mechanism of action of each DDR inhibitor, and these alterations are particularly prominent after irradiation. In all cases, inhibitors are capable of preventing radiation induced expression of products (olaparib) or phosphorylation of target proteins (AZD0156, AZD6738)

Point 2:

The second comment highlighted from the previous review is about the lack of mechanistic evidence supporting the differential response of the two cell lines to the combination of the inhibitors plus single and fractionated radiation doses. I agree with the authors that this would require additional experimentation, although this is not completely beyond scope. Based on the results provided, it is therefore difficult to predict based on just the two cell lines utilised generating different responses, how broadly applicable and effective the strategy of DDR inhibition with fractionated radiation in in vitro breast cancer models is and what are the key driving factors for this response. In this respect, the paper is limited, although this does provides some interesting observations.

We acknowledge that the current work is unfortunately limited in that it does not identify the key mechanism for this elevated sensitivity to fractionated radiotherapy in the presence of DDR inhibitors. We have investigated the key mechanisms associated with varying radiosensitivity (DNA damage repair, cell cycle dysregulation) and while there is some impact on DNA repair, this does not seem to be sufficient to fully explain the impact on fractionated responses. Further investigations across other pathways and mechanisms, while of interest, would be increasingly speculative, and it is unclear how much additional data would be required to resolve these issues.

Despite this, we feel that the current manuscript is still valuable, as it clearly identifies a novel interaction between fractionation and DNA repair inhibitors which does not seem to follow the standard predictions from the LQ model. Even if only empirically identified at present this is still potentially significant for the design of future studies combining DNA repair inhibitors and radiotherapy, and it provides a foundation for future work in this area to better understand the interaction between these different agents.

Reviewer 2 Report

The authors responded to reviewers' comments promptly. Only criticism from me is that, in figure 11, changes in the phosphorylation of CHK2 and CHK1 as well as expression of PAR are descriptive and subjective. More objective presentation, for example, with band density would be preferable.

Author Response

We would like to thank the reviewer for their positive feedback on the manuscript, and their further useful suggestions. We would also like to apologise for the delay in responding to these requests, as radiation work in our lab was temporarily delayed due to equipment failures. These have now been resolved, and we have revised the manuscript with respect to these suggestions, as described below.

Point 1:

The authors responded to reviewers' comments promptly. Only criticism from me is that, in figure 11, changes in the phosphorylation of CHK2 and CHK1 as well as expression of PAR are descriptive and subjective. More objective presentation, for example, with band density would be preferable.

In response to the reviewer’s suggestion, we have performed Western blots using pATM, pATR, and PAR as target proteins of AZD0156, AZD6738 and olaparib, respectively. Results are shown in the new Figure 10, a new supplementary figure, and further described in Section 3.7. The blots were also quantified to show the relative degree of inhibition graphically.

Changes in expression or phosphorylation of key targets are in agreement with the mechanism of action of each DDR inhibitor, and these alterations are particularly prominent after irradiation. In all cases, inhibitors are capable of preventing radiation induced expression of products (olaparib) or phosphorylation of target proteins (AZD0156, AZD6738). In addition, in most cases the total level of expression/phosphorylation is also significantly reduced. Full abrogation of phosphorylation was not achieved in most cells, likely due to the importance of these proteins meaning that completely removing their function was too toxic.

Round 2

Reviewer 1 Report

I accept the authors revisions and responses to my comments. A minor suggestion is that the new data in Figure 10 should be normalised (control set to 1.0), as the numbers are meaningless how they are currently presented.

Author Response

Thank you for the feedback. As suggested, we have normalised the protein ratios in the western blots (Figure 10 & Supplementary Figure 1) to the untreated control cells, and the manuscript has been updated accordingly. 

This manuscript is a resubmission of an earlier submission. The following is a list of the peer review reports and author responses from that submission.

Round 1

Reviewer 1 Report

In this study, the impact of the DNA damage response (DDR) inhibitors targeting ATM, ATR and PARP on the radiosensitisation of two breast cancer cell lines has been investigated, using both single dose and fractionated (two equal doses, 24 h apart) x-ray radiation. Largely using clonogenic assays, the major conclusion is that the DDR inhibitors lead to radiosensitisation of both MDA-MB-231 and MCF-7 cells following fractionated radiation, but only ATM inhibition is effective following single dose radiation (majorly in MDA-MB-231 cells). Additional data shown through 53BP1 foci analysis demonstrates that the DDR inhibitors delay repair of DNA double strand breaks after a single radiation dose, whereas there appeared to be minimal impact on cell cycle distribution.

Overall, this is presenting interesting and novel data examining DDR inhibition following single and fractionated radiation in vitro. Despite this, there are a number of critical comments and additional experimentation that is required prior to publication.

Specific comments

  1. Whilst IC50 values of the DDR inhibitors have been determined (Table 1), the effectiveness of the drugs in inhibiting the respective target (ATM, ATR and PARP) post-irradiation should be shown to confirm that the concentrations used in clonogenic assays are effective. Therefore, phosphorylation of ATM and ATR, as well as poly(ADP-ribose) polymer formation should be shown by immunoblotting post-irradiation in the presence versus the absence of the respective inhibitors.
  2. Given that the fractionated experiments involve radiation doses 24 h apart, it is unclear in experiments involving DDR inhibition (e.g. Figures 6-8) whether cells were treated with the drugs at both time points 1 h pre-irradiation, or whether a single drug dose was used. Despite this and similar to the comment above, effective inhibition of the enzyme target (ATM, ATR and PARP) needs to be shown with the dosing strategy used in these fractionated experiments (in addition to the single dosing).
  3. An important comment is that the data clearly point to a differential effect of the inhibitors in the two breast cancer cells used in the study, however there is no mechanistic evidence to understand why this is the case. Specifically, MDA-MB-231 are highly sensitive to ATM inhibition with single dose irradiation, whereas MCF-7 are insensitive to all inhibitors. In contrast following fractionated radiation, MDA-MB-231 are again sensitive to ATM inhibition but to a lesser degree PARP inhibition, whereas MCF-7 cells are equally sensitive to ATM, ATR and PARP inhibition. The authors in the Discussion refer to differential levels and reliance on ATM possibly being an important factor in response to AZD0156 in MDA-MB-231, however more mechanistic evidence is required to support the different cellular responses and apparent reliance on the DDR pathways.
  4. It is unclear why the authors chose 53BP1 as a marker of DNA double strand break repair (Figure 9), rather than γH2AX which is typically used. However, and specifically in response to ATR inhibition, the impact on RAD51 foci should be analysed. I would also recommend that these data are supplemented with direct analysis of DNA double strand breaks (e.g. comet assay) rather than just surrogate markers. Furthermore, this data only correlates with the relevant clonogenic assay using single dose radiation, whereas the impact on DNA double strand break levels and repair should also be analysed following fractionated radiation. This is also important as to demonstrate the continued effectiveness of the DDR inhibitors using this strategy (see point 2).
  5. Interestingly, there is no significant evidence that the DDR inhibitors have an impact on cell cycle distribution of the breast cancer cells post-irradiation (Figure 10). However, it is noticeable that a relatively low dose of radiation (2 Gy) is used in these experiments, and which does not appear to activate significant cell cycle arrest. If the aim is to prove that DDR inhibitors cause deficiencies in checkpoint activation, then these experiments need to be repeated using a higher dose of radiation. As in point 4 above, there is an argument that cell cycle analysis should be completed using single and fractionated radiation doses.
  6. In terms of more minor comments, the Discussion is quite extensive and speculative in places. If some of the experiments suggested above are added, then this section could be shortened and more refined. Additionally, the dose response curves relating to the IC50 data of the DDR inhibitors shown in Table 1, should be added at least in the Supplementary Data.

Reviewer 2 Report

In the manuscript “DNA Repair Inhibitors Potentiate Fractionated Radiotherapy More Than Single-Dose Radiotherapy in Breast Cancer Cells”, the authors investigated radiosensitizing effect of three pharmacological inhibitors of DNA damage response in in vitro setting. The merit of this study is that they evaluated effect of these inhibitors not only on single dose irradiation but also on fractionated irradiation. They found radiosensitizing effect of ATR inhibitor AZD6738 and PARP inhibitor olaparib only on the fractionated irradiation while they found radiosensitizing effect of ATM inhibitor AZD0156 both on single dose and fractionated irradiation to argue importance of irradiation protocol in in vitro study.

Overall, this study is well organized, and manuscript is well written with detailed introduction. I noticed numbers of points which require revision as listed below.

  1. In the section 3.7, the authors found no difference in the cell cycle distribution in the presence of DDR inhibitors, concluding that cell cycle modifications do not contribute to radiosensitization by AZD0156, AZD6738 and Olaparib. However, in the irradiation dose they used in this experiment (2Gy), there was little change in cell cycle distribution in the first place. It would be important to examine effect of DDR inhibitors on reversal of cell cycle arrest induced by IR. The reason of selecting this irradiation dose should be explained or different dose of IR should be tested.

  1. In the section 3.6, the authors found delay of DSB repair in the presence of not only ATM inhibitor but also of ATR inhibitor and PARP inhibitor. They also discuss about residual DNA repair in the presence of DDR inhibitors, reasoning to the presence of multiple mechanisms of DNA repair (line 477). Is it possible that residual DNA repair in the presence of ATM inhibitor depends on ATM or PARP? It would be informative to see effect of multiple DDR inhibitors in combination.
